# Antimicrobial Peptides and Their Biomedical Applications: A Review

**DOI:** 10.3390/antibiotics13090794

**Published:** 2024-08-23

**Authors:** Ki Ha Min, Koung Hee Kim, Mi-Ran Ki, Seung Pil Pack

**Affiliations:** 1Institute of Industrial Technology, Korea University, Sejong-Ro 2511, Sejong 30019, Republic of Korea; alsrlgk@gmail.com (K.H.M.); allheart@korea.ac.kr (M.-R.K.); 2Department of Biotechnology and Bioinformatics, Korea University, Sejong-Ro 2511, Sejong 30019, Republic of Korea; wood1018@korea.ac.kr

**Keywords:** antimicrobial peptides, antibiotic alternatives, antimicrobial agents, antimicrobial resistance, biomedical applications

## Abstract

The emergence of drug resistance genes and the detrimental health effects caused by the overuse of antibiotics are increasingly prominent problems. There is an urgent need for effective strategies to antibiotics or antimicrobial resistance in the fields of biomedicine and therapeutics. The pathogen-killing ability of antimicrobial peptides (AMPs) is linked to their structure and physicochemical properties, including their conformation, electrical charges, hydrophilicity, and hydrophobicity. AMPs are a form of innate immune protection found in all life forms. A key aspect of the application of AMPs involves their potential to combat emerging antibiotic resistance; certain AMPs are effective against resistant microbial strains and can be modified through peptide engineering. This review summarizes the various strategies used to tackle antibiotic resistance, with a particular focus on the role of AMPs as effective antibiotic agents that enhance the host’s immunological functions. Most of the recent studies on the properties and impregnation methods of AMPs, along with their biomedical applications, are discussed. This review provides researchers with insights into the latest advancements in AMP research, highlighting compelling evidence for the effectiveness of AMPs as antimicrobial agents.

## 1. Introduction

The increasing prevalence of antibiotic resistance and the shortage of novel antibacterial drugs have critically endangered medical care by exposing patients to severe infections. These resistant infections often lead to higher mortality rates due to the lack of effective treatments, necessitating longer hospital stays, intensive care, and more expensive medications [1]. To combat antimicrobial resistance, a multi-faceted approach is essential. This includes the prudent use of antibiotics, enhanced infection prevention and control measures, and the development of new antimicrobial agents. In the search for innovative antimicrobial strategies, antimicrobial peptides (AMPs) have demonstrated significant potential. AMPs are widely found in nature and play a crucial role in the innate immune systems of various organisms [2], exhibiting broad inhibitory effects against bacteria, fungi, parasites, and viruses [3,4,5]. AMPs are resilient to extremes of pH and temperature, yet they are vulnerable to proteases in the host’s small intestine [6,7]. They help modulate the host’s natural defense systems by preventing the adhesion and colonization of harmful bacteria [8,9]. The term AMP encompasses peptides from all organisms and microorganisms (eukaryotes and prokaryotes) and from two biosynthetic pathways (ribosome-independent and ribosome-dependent). Those AMPs derived from higher organisms are often referred to as host defense peptides (HDPs) [10,11,12,13]. Recent research indicates that AMPs produced by bacteria or that are modified post-translationally, such as bacteriocins, are more effective in suppressing harmful bacterial colonization compared to those produced by eukaryotes [14].

AMPs typically consist of 5 to 100 amino acid residues, with 97% having 12 to 50 residues and an average length of 28 [2,15]. Features of peptides with antimicrobial activity are categorized based on charge, hydrophobicity, amphipathicity, and secondary structure (Figure 1) [16,17]. Almost all AMPs (96%) have an average positive charge of +4 to 5 mV, which benefits attachment to negatively charged components of bacterial surface, including lipopolysaccharide (LPS), lipoteichoic acid, and mannoproteins [18,19]. The positive net charge of the AMPs is impacted by the amounts of arginine (Arg), lysine (Lys), and histidine (His) in the sequence. Anionic AMP also exists in the human innate immune system, but the mechanism is still unknown [17]. Another essential factor for antibacterial activity is hydrophobicity in AMPs, commonly described as the proportion of hydrophobic residues in the peptide, which influences the distribution of AMPs in the hydrophobic part of the bacterial membrane [20,21]. Upon comparing and analyzing numerous AMPs, it has been determined that hydrophobic amino acid residues make up approximately 40 to 60% of the total number of amino acid residues [22]. The secondary structure of peptides is also important for antibacterial activity. Despite an unordered form in an aqueous solution, AMPs can interact with phospholipids and then change to a stable structure [19]. Finally, amphipathicity, the separated ratio of hydrophilic and hydrophobic parts, is related to attachment, interaction, and penetration of the membrane [23]. Considering these factors, there are efforts to optimize or newly synthesize AMPs to enhance their antibacterial effects and reduce their toxicity [24,25].

AMPs have been extensively investigated as to their mechanisms of action and potential therapeutic applications (Figure 2) [26]. Current clinical studies predominantly focus on their antimicrobial properties and the feasibility of topical administration. However, recent reports indicate that AMPs also possess promising attributes for use in wound healing, cancer therapies, and potentially as novel cosmetic ingredients [27,28,29]. Multiple studies have reported that AMPs suppress the growth of pathogens and strengthen the immunological functions of the host [30]. AMPs kill pathogens through various mechanisms, such as binding to peptidoglycans to destroy cell walls, linking to phospholipids to perforate cell membranes, interfering with RNA reverse transcription, and activating host immune systems [31]. Recent research has demonstrated that AMPs, especially bacteriocins, can regulate the gut microbiota to secrete quorum-sensing signaling molecules. These molecules suppress endotoxin production from pathogens and activate the mammalian target of rapamycin (mTOR) pathways in intestinal epithelial cells to enhance intestinal barrier function [32,33]. Importantly, the extensive application of AMPs is unlikely to produce resistance genes in gut bacteria [34,35], making them one of the most effective substitutes for antibiotics in human biomedicine.

Despite their promise, AMPs face several limitations for clinical application. Natural peptides often exhibit instability in the gastrointestinal tract and other bodily fluids, along with poor absorption, distribution, and rapid metabolic degradation and excretion, resulting in limited bioavailability [36,37]. Their flexible structures also raise concerns about possible interactions with unintended components, potentially leading to adverse effects. Furthermore, according to the data repository of antimicrobial peptides (DRAMP) database, approximately 67% of known AMPs from various sources, particularly 78% from human sources, are greater than 20 amino acid residues in length [38,39,40,41,42], with many containing dominant residues as cationic residues and hydrophobic residues [42]. These characteristics, including length and composition, pose challenges and costs in their synthesis and application.

To overcome these challenges, recent studies have proposed both chemical and bioengineering strategies aimed at developing peptide formulations that are more potent, selective, and metabolically stable, while also being cost-effective and less likely to induce undesired side effects. These innovative approaches, coupled with new discoveries regarding their biological roles, have propelled AMPs into an emerging category for clinical applications [43,44,45]. In essence, AMPs are a promising option for creating alternatives to antibiotics, offering effective solutions for various uses in human health, animal production, and disease control.

This review offers a comprehensive examination of AMPs as an alternative to traditional antibiotics. It examines their antimicrobial mechanisms, immunomodulatory properties, and strategies for enhancing their effectiveness. Moreover, it addresses the potential of AMPs in biomaterials for biomedical applications and discusses the current challenges and future prospects in the development of AMPs.

## 2. Modes of Action and Mechanisms of AMPs

Most AMPs exert their effects through multiple mechanisms, primarily by targeting bacterial cell membranes (Figure 3). AMPs can bind to and penetrate the bacterial membrane and then interact with the phospholipid components of the cytoplasmic membrane, leading to pore formation, which disrupts membrane integrity, causing leakage of cellular contents and leading to cell lysis and death [3]. In addition to disrupting membranes, AMPs can also target essential cellular mechanisms within pathogens through binding to DNA, RNA, and protein and inhibiting functions of them, further contributing to their destruction [46]. Furthermore, AMPs have immunomodulatory properties, activating interleukins, chemokines, and cytokines to enhance the immune response and aid in clearing pathogens from the body [47]. While the exact mechanisms are not fully understood, it is evident that AMPs can impair nucleic acid biosynthesis, translation, cell division, and cell wall biosynthesis, or even induce apoptosis without necessarily compromising membrane stability [48]. In addition to their direct antimicrobial actions, AMPs inhibit biofilm formation by disrupting microbial adhesion and interfering with quorum sensing, while also degrading the extracellular matrix of existing biofilms, making them more susceptible to treatment [49,50]. These actions can enhance their effectiveness against infections involving biofilms.

### 2.1. Membrane Targeting

AMPs utilize a cell membrane-targeting mechanism wherein positively charged and amphiphilic AMPs bind to negatively charged amphipathic phospholipids in the cell membrane. This binding results in the formation of pores and channels within the membrane, thereby disrupting its integrity and function [48,52,53,54,55]. To date, five models describing the modes of action of AMPs on cell membranes have been identified: the barrel-stave model, the toroidal pore model, the carpet model, the aggregation channel model, and the hydrological mechanism model [52,56,57]. These models are illustrated in Figure 4. 

The barrel-stave model describes the formation of pores that result in cytoplasmic outflow, membrane collapse, increased permeability, and cell death [48,53,57] (Figure 4). AMPs that employ this mechanism include ceratotoxins, protegrins, and alamethicin. In this mechanism, the AMPs insert themselves perpendicularly into the lipid bilayer, forming a transmembrane channel lined with the peptides’ hydrophilic regions [53,57].

In the toroidal pore model, AMPs interact with lipid head groups, inducing a curvature in the lipid bilayer that allows the peptides to insert into the membrane and form continuous pores and channels through it. This bending of the membrane bilayer facilitates the formation of a pore lined with both peptides and lipid head groups. Representative AMPs that utilize this mechanism include magainin2, protegrins, actinoporins, and melittin [48,53,57].

The carpet model describes a mechanism where AMPs cover the cell membrane surface in a manner akin to a carpet, interacting extensively with the phospholipid head groups. This widespread interaction destabilizes the membrane, leading to high local concentrations of peptides that subsequently cause the disintegration and permeabilization of the phospholipid bilayer. Examples of AMPs that operate via this model include cecropins, cathelicidin LL-37, and indolicidin [48,53,57].

In the aggregate channel model, AMPs spontaneously form unstructured aggregates around the pathogen’s membrane. These aggregates induce the formation of channels within the membrane, resulting in the leakage of cytoplasmic contents and compromising the cell’s integrity [56].

The floodgate mechanism, a recently proposed model, suggests that during the initial phase of AMP attack, α-helical peptides form transient toroidal gaps in the pathogen’s cell membrane. This mechanism posits that the AMPs initially apply stress to the membrane through combined hydrophobic and electrostatic interactions. Following this initial perturbation, the peptides recruit nearby unbound AMPs to amplify the membrane disruption, leading to extensive membrane damage and increased permeability [58].

**Figure 4 antibiotics-13-00794-f004:**
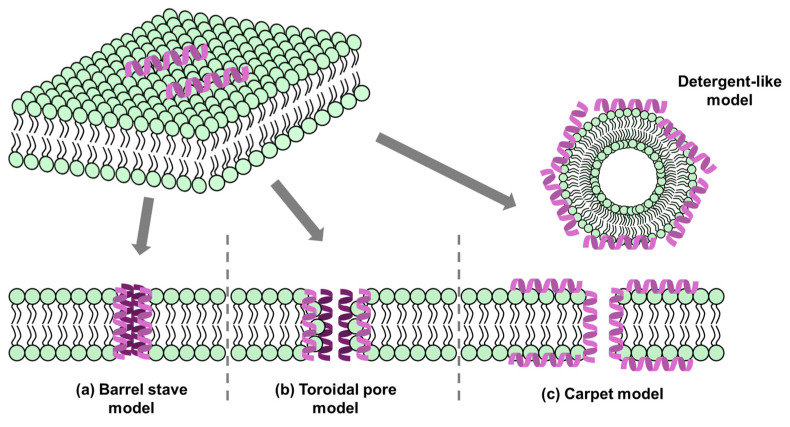
Models of action for extracellular AMP activity. (**a**) Barrel-stave model: AMPs aggregate into multimers and insert into the lipid bilayer, aligning parallel to the phospholipids and forming transmembrane channels. (**b**) Toroidal pore Model: AMPs embed perpendicularly into the membrane, bending to form a continuous pore through the lipid bilayer. (**c**) Carpet model: AMPs accumulate on the membrane surface, acting similarly to detergents, leading to membrane disruption and cell lysis [59].

### 2.2. Non-Membrane Targeting

AMPs can enter cells either through direct penetration or endocytosis. Once inside the cytoplasm, they identify and act on specific targets and are classified into categories based on these targets. Some AMPs directly penetrate the cell membrane, allowing them to act on vital bacterial organelles and intracellular proteins, or target RNA, DNA, and protein synthesis [54,57,60].

AMPs interfere with transcription, translation, and molecular chaperone folding by targeting related enzymes and effector molecules. For example, Bac7(1–35) targets ribosomes to inhibit protein translation [61], while Tur1A inhibits protein synthesis in *Escherichia coli* and *Thermus thermophilus* by blocking the transition from the initiation to the elongation step, though they bind to ribosomes and interact with the ribosomal peptide exit tunnel in different ways [62]. Additionally, genome-wide transcriptional analysis shows that the AMP DM3 affects several key intracellular pathways involved in protein biosynthesis [63]. Chaperones, crucial for folding and assembling newly synthesized proteins into their functional stereoisomeric forms, contribute to the cell selectivity of AMPs and reduce cytotoxicity. For instance, pyrrhocoricin and drosocin prevent DnaK from refolding misfolded proteins by permanently closing the DnaK peptide-binding cavity [54,64,65].

AMPs inhibit nucleic acid biosynthesis by targeting key enzymes or degrading nucleic acid molecules. For instance, indolicidin, a C-terminal-amidated cationic Trp-rich AMP with 13 amino acids, specifically targets the abasic site of DNA, crosslinking single- or double-stranded DNA and inhibiting DNA topoisomerase I [66]. Similarly, TFP (Tissue factor pathway inhibitor) 1-1TC24, an AMP from tongues, enters the cytoplasm of target cells after membrane rupture and degrades DNA and RNA [67]. Moreover, AMPs can bind to nucleic acids and proteins, disrupting their conformation to inhibit synthesis. This mechanism is common in histone-derived AMPs, such as buforin II and indolicidin, which are effective against both Gram-negative and Gram-positive pathogens.

AMPs inhibit various metabolic activities, including protease activity and cell division. For example, histatin 5 inhibits host and bacterial proteases, while eNAP-2 and indolicidin target microbial serine proteases, elastase, and chymotrypsin [54]. Cathelicidin-BF, from *Bungarus fasciatus* venom, blocks thrombin-induced platelet aggregation and protease-activated receptor 4 [68]. AMPs also inhibit cell division by disrupting DNA replication, cell cycle progression, and chromosome separation [69]. APP kills *C. albicans* by inducing S-phase arrest, and MciZ inhibits bacterial cell division by affecting Z-ring formation [70]. Additionally, some AMPs, such as histatin 5, damage fungal organelles by interacting with mitochondria to produce ROS and induce cell death [71].

AMPs also possess a wide range of immunomodulatory properties against various infections, including the induction of cytokine production, chemoattraction, and immune cell activity [72,73,74]. LL-37, the only human member of the cathelicidin family, and Human β-defensin stimulate the p38 and ERK1/2 MAPK pathways in keratinocytes and then induce the secretion of interleukine-18 (IL-18) [75]. Human β-defensin-3 can bind to CCR6 and then activate caspase, mitogen-activated protein kinase (MAPK), and nuclear factor-kappa B (NF-kB) pathway, resulting in the secretion of IL-37 [76]. Human lactoferrin-derived peptide 1–11 indirectly promotes T helper 17 cell polarization by inducing monocyte-dendritic cell differentiation [77]. Green-fluorescent protein-tagged Retrocyclins and Protegrin-1 induce calcium mobilization and degranulation of human mast cells by Mas-related G protein coupled receptor X2 independently of formyl peptide receptor-like 1 as an AMP receptor [78].

## 3. AMP Modification to Enhance Antimicrobial Activity

AMPs are a suitable candidate for counteracting antibiotic resistance because they exhibit broad antibiotic activity. Peptide sequences based on natural AMPs are designed and optimized for higher antibacterial activity and safety [79]. Prior to discussing recent advancements in the combination of component materials for AMPs, this section first presents the latest research trends focused on the design of AMPs. Specifically, Section 3 addresses the enhancement of antimicrobial activity through peptide modification strategies. Although recent advancements in the composition of AMPs are significant, the latest research trends in AMP design are equally diverse, with all approaches focused on optimizing the utilization of AMPs. Recently, various biomaterials conjugated with AMPs were found to induce selectivity between bacteria and mammalian cells, stability to enzymes, and high penetration rates [80] (Table 1).

Yu et al. attached mPEG (1000 Da) to the AMP T9W at the N-terminal or C-terminal ends (CT9W1000), respectively [81]. CT9W1000 formed self-assembled nanoparticles, while N-terminally modified T9W showed a nanowire structure. Neither T9W nor CT9W1000 demonstrated toxicity to IPEC-J2, RAW 264.7, or HEK 293 T cells at 4–256 μg/mL. CT9W1000 exhibited enhanced antibacterial effects against Gram-negative and Gram-positive bacteria compared to T9W, while the N-terminally modified T9W lost antibacterial activity. In addition, PEGlyation of the peptide provided resistance to salt, serum, and trypsin, which produced greater stability and therapeutic effects. In mice infected with *Pseudomonas aeruginosa* (*P. aeruginosa*), CT9W1000 decreased bacterial burden and reduced lung tissue injuries. The low-molecular-weight PEG demonstrated a synergistic antibacterial effect through a positively charged micelle formulation, which facilitated the absorption and attack of a negatively charged bacterial membrane. This can be attributed to the antibacterial activity observed against a diverse range of bacterial strains.

Song et al. developed structurally nanoengineered antimicrobial peptide polymers (SNAPPs) using polymerization of lysine and valine N-carboxyanhydride monomers initiated from poly(amidoamine) dendrimers [82]. SNAPPs were used to create a nano-thickness (1.8–1.9 μm diameter) capsule by interacting with tannic acid or the metal phenolic network. The microcapsule showed sustained release (~40%) at pH 7.4 after 160 h and maintained structure after the release of SNAPP. The minimum inhibitory concentration (MIC) of capsules was 29.3 μg/mL against *E. coli*, an improved value compared with other antimicrobial materials (>37 μg/mL). In addition, the microcapsule remains stable after the nebulization process and has higher colocalization with endo/lysosomes for targeting lung infections and diseases. Modification of AMP in the form of SNAPP can increase the density of AMP in one space compared to free AMP and then bacterial membrane depolarization, which leads to enhanced antibacterial activity [83].

Patrulea et al. conjugated the AMP dendrimer G3KL and chitosan derivates using covalent crosslinking [84]. Various sulfo-crosslinkers were used and compared for conjugation between G3KL-cysteine and chitosan derivates. These crosslinkers were then optimized to enhance MIC and solubility. Biopolymer modification improved the antibacterial activity of the G3KL-embedded biopolymer matrix at least 20-fold compared to just physically mixing the AMP with biopolymer. The conjugate showed stable bacterial killing activity against *P. aeruginosa* even after serum incubation and sustained bacterial growth inhibition effect for 48 h, while non-modified G3KL needed twice the MIC after 6 h. Furthermore, the GSKL with chitosan had reduced cytotoxicity against red blood cells and human dermal fibroblasts. By linking chitosan, the protein degradation resistance of AMP dendrimer was improved, thereby enhancing its stability. In addition, it can be attributed to the more robust interaction with intracellular cytoplasmic components through enhanced membrane translocation by better attachment to bacterial membranes by chitosan.

Modifications to antimicrobial peptides (AMPs) have shown considerable promise in enhancing their antimicrobial properties. Key strategies include chemical modifications, such as amino acid substitution and cyclization, which improve the peptides’ stability and activity. Structural optimizations, including increased helicity and adjusted amphipathicity, further boost antimicrobial efficacy. Despite these advancements, challenges remain. Modified AMPs may exhibit non-specific toxicity to host cells, and there is a risk of microbial resistance. Additionally, the complexity and cost of synthesis, stability concerns, and potential immunogenic responses are significant obstacles. Addressing these issues is crucial for advancing the clinical application of AMP-based therapies.

**Table 1 antibiotics-13-00794-t001:** Biomedical applications of AMPs modified by various materials.

AMP	Materials	Biomedical Results	Refs.
T9W	Poly (ethylene glycol)	Anti-tripsin ability by self-assembled micelle structure Enhanced antibacterial effect 1.5–4 times against Gram-negative and Gram-positive bacteria Reduced lung injury and pro-inflammatory cytokines by *P. aeruginosa* PAO1	[81]
Lysine and Valine	Poly (amidoamine)	Stable aerosolization, sustained release by nanoparticle formulation with tannic acid and iron ion Stability in intracellular environment Colocalization with endo/lysosome expected targeting bacteria in lung macrophage	[82]
KR-12	Poly carprolactone Poly(ethylene glycol) methyl ether methacrylate	Specific binding effects by coating with macropharge membrane Antibacterial and antibiofilm effects against *E. coli*, *S. aureus*, and MRSA Increased adhesion and protection effects of main organ-involved sepsis	[85,86]
KRWWKWWRR	Hydoxyapatite binding peptide-1	Providing implant with inhibited adhesion and antibacterial effects against *E. coli* and *S. mutans*	[87]
9P2-2	Ampicillin	Improved antibacterial activity against ampicillin-resistant *A. baumannii* Non-toxicity to mammalian cell	[88]
WRK	Acrylamide	High antibacterial activity by bacteria-mediated polymerization against *E. coli* and *P. aeruginosa*	[89]
G3KL	Chitosan	Boosting bacterial killing (*P. aeruginosa*) Decreased hemolytic activity by conjugation High biocomaptibility of dressing form	[84]
LWFYTMWH	Poly (ethylene glycol)	Antibacterial activity against *E. coli* and *Bacillus* sp.	[90]
Chimeric PR-39	Cell penetrating peptide (R6)	Fast and non-resistant antibacterial effect High biosafety in vitro (64 uM of peptide) and in mice (~30 mg peptide/kg)	[91]

*Escherichia coli* (*E. coli*). *Staphylococcus aureus* (*S. aureus*). Methicillin-resistant *Staphylococcus aureus* (MRSA). *Streptococcus mutans* (*S. mutans*). *Acinetobacter baumannii* (*A. baumannii*). *Pseudomonas aeruginosa* (*P. aeruginosa*).

## 4. AMP Application for Biomaterials

To effectively utilize AMPs in biomedical applications, several biomaterial strategies have been developed. These strategies aim to enhance the stability, efficacy, and delivery of AMPs to target sites [92,93,94,95]. The main approaches include coating medical devices with AMPs [17,96], modifying AMP structures for improved performance [97], and exploring other innovative strategies to maximize their therapeutic potential [96,98]. Each of these methods comes with its own set of challenges and advantages, discussed in the following sections.

### 4.1. Surface-Based Applications

The development of multifunctional smart coatings with versatile properties is garnering significant attention, driven by the escalating demand for advanced medical and industrial applications [96,97,99,100,101]. These coatings are engineered to tackle various challenges, notably antimicrobial resistance and the efficient delivery of therapeutics [96,102,103] (Table 2).

Dai et al. synthesized a self-assembling nanofibrillar membrane composed of antimicrobial GL13K and collagen that promotes antimicrobial activity and enhances bone formation in vivo [104]. GL13K nanofibrils substantially increased hydrophobicity, reduced biodegradation, and stiffened the collagen matrix, leading to antimicrobial and antifouling activity against *S. gordonii*. In collagen membranes coated with GL13K, most bacteria had compromised membranes, while in control groups without GL13K, most bacteria remained viable. The hydrophobicity and contact killing effect of a membrane contribute to its antimicrobial properties. Inclusion of GL13K did not affect fibroblast proliferation or pre-osteoblast differentiation, and bone formation in a rat calvarial model was remarkably accelerated compared to a gold-standard collagen membrane. 

Li et al. investigated the immobilization of the GL13K peptide onto sandblasted and acid-etched (SLA)-treated titanium [105]. The AMP GL13K was successfully attached to SLA-treated titanium surfaces using KH-550 as a silane coupling agent. A certain amount of GL13K peptide was covalently immobilized onto the titanium surface using KH-550, while another portion was physically deposited and adsorbed. This coating showed sustained release of GL13K, providing effective antibacterial properties against *E. coli*, *S. aureus*, and drug-resistant methicillin-resistant *Staphylococcus aureus* (MRSA). SLA-treated samples not only promoted the adhesion of osteoblasts, but also facilitated bacterial adhesion. However, after coating with GL13K, the samples exhibited significant antibacterial activity and good cytocompatibility.

Gao et al. investigated the synergistic effect of two commercial AMPs in a blown extrusion fabrication process [106]. Starch/Poly (butylene adipate-co-terephthalate) films incorporating two AMPs (nisin and ε-PL) were successfully developed using extrusion blowing. Starch/polybutylene adipate-co-terephthalate (PBAT) composite films exhibit exceptional oxygen barrier properties and demonstrate considerable potential as carriers for functional components. Consequently, these starch/PBAT composites present a promising matrix for the development of antimicrobial food packaging films. Adding either nisin or ε-PL alone resulted in heterogeneous microstructures and decreased the mechanical properties of the films. However, the SP-PN1/2 film containing both peptides exhibited superior tensile strength, flexibility, and a more uniform morphology. These films showed higher moisture permeability and improved oxygen barrier properties. The combination of ε-PL and nisin demonstrated synergistic antimicrobial effects, effectively inhibiting over 90% of the growth of foodborne pathogens (*S. aureus* and *E. coli*) and extending the shelf life of fresh peaches. 

Current trends include the incorporation of AMPs into coatings and films for medical devices, such as catheters and implants, to prevent bacterial colonization and biofilm formation. However, several limitations challenge the broader application of AMP-based surface technologies. The durability and stability of AMP coatings can be compromised by mechanical wear and environmental factors, leading to diminished efficacy over time. High production costs and scalability issues remain significant barriers to widespread adoption. Additionally, potential toxicity to human cells and environmental impacts require thorough evaluation to ensure safety.

AMPs offer significant advantages in surface-based applications, particularly for medical devices. They effectively prevent bacterial adhesion and biofilm formation, as demonstrated by GL13K-coated collagen membranes showing strong antimicrobial and antifouling properties against *S. gordonii* [107]. These coatings are also biocompatible, promoting cell proliferation and differentiation without adverse effects. The multifunctional nature of AMP coatings addresses challenges such as antimicrobial resistance and therapeutic delivery. However, their durability and stability can be compromised over time due to mechanical wear and environmental factors. High production costs and scalability issues present significant barriers to widespread adoption, and potential toxicity to human cells and environmental impact require thorough evaluation.

**Table 2 antibiotics-13-00794-t002:** Biomedical applications of AMP-coated materials.

AMP	Materials	Biomedical Results	Ref.
GL13K	Collagen membrane	Improved antimicrobial and antifouling activity, accelerated bone formation	[104]
GL13K	Mineralized collagen gel	Killing of Gram-negative *E. coli* and Gram-positive *S. Gordonii*, cytocompatible with human bone marrow-derived mesenchymal stromal cells	[107]
GL13K	Sandblasting and acid-etching-treated titanium	Sustained-release property, antibacterial property, oestoblast proliferation, and adhesion in vitro	[105]
QAGSNKGASQKGMS	Dopamine, 304 stainless steel	Antifouling capacity, antibacterial and antialgal properties, superior anticorrosion	[108]
ε-Polylysine hydrochloride (ε-PL), Nisin,	Starch/PBAT film	Higher moisture permeability and oxygen barrier property, synergistic antimicrobial effect	[106]
Lysozyme	Doapmine-modified graphene oxide	Antimicrobial activity, accelerated wound closure, reduced inflammation, improved angiogenesis, and accelerated re-epithelialization	[109]
Hs05 and Hs06	Ureasil–polyether hybrid polymeric films	Antimicrobial activity	[110]
TCP-25	Polyurethane	Anti-infective and anti-inflammatory effects in vitro and in vivo, reduced the concomitant inflammatory response	[111]
M2-DA	Stainless steel	Excellent antibacterial activity	[112]
HHC-36	Pectolite nanorods on Ti implants	Antimicrobial activity while promoting cell adhesion, regulates the degradation of Ca- and Si-based ceramic	[113]
RRP9W4N	Mesoporous titania	Antibacterial activity, no negative effects on in vivo osseointegration	[114]
RRP9W4N	Surface-modified titanium implants with elastin-like polypeptide	Antibacterial activity, enabled mammalian osteogenic cell adhesion	[115]
HX-12C	Chitosan flim	Good antibacterial actibity, strong antibiofilm ability	[116]

### 4.2. Particle-Based Applications

The combination of nanoparticles and antibacterial peptides not only overcomes antibiotic resistance, but also brings about a synergistic effect by combining various functionalities. These modifications have helped to overcome limitations such as toxicity and boost target effects (Table 3).

Rajchakit et al. investigated one-pot synthesis of size-controlled (10 nm) gold nanoparticles (NPs) selectively conjugated to lipopeptides and determined their antibacterial activity [117]. The study highlights the development and efficacy of peptide-AuNP conjugates derived from a linear analogue of the antimicrobial lipopeptide battacin. These conjugates exhibit potent antimicrobial activity against a range of Gram-positive and Gram-negative bacteria, with MICs between 0.13 and 1.25 μM, respectively. They also show significant antibiofilm activity, achieving 90% inhibition of initial biofilms and an 80% reduction in preformed biofilms at low micromolar concentrations. The conjugates are stable in rat serum and non-toxic to mammalian cell lines in vitro (≤64 μM) and in vivo (≤100 μM). Additionally, these conjugates display faster bacterial killing kinetics compared to the free peptide, are neither hemolytic nor cytotoxic up to 50 times their MIC, and possess improved serum stability. These enhanced antibacterial effects of the conjugate compared to the free form are mainly due to the increased AMP density and surface area by the formation of AMP-NP. It allows the conjugate NP to bind to the bacterial membrane, depolarize it more strongly, and penetrate it, thereby bursting the membrane. The study also addresses concerns regarding NP size-related toxicity, noting that NPs below 10 nm are generally more toxic to mammalian cells. The size-controlled synthesis method for NPs and their conjugation to AMPs presents potential new applications in biomedical nanoscience.

Liu et al. attached the AMP cathelicidin-BF to the surface of epigallocatechin-3-gallate (EGCG)-loaded silk fibroin nanoparticles (CBF-EGCG NPs) for therapeutic application against ulcerative colitis [118]. The negatively charged EGCG-NPs were neutralized by the positively charged CBF, thus resulting in the uniform formation of CBF-EGCG NPs at 190 nm. The CBF-EGCG NPs showed non-cytotoxicity against CT-26 cells and RAW 264.7 macrophages during co-incubation at a range of 1.25–50 μg/mL. Interestingly, these NPs were phagocytosed into the cell at a higher level in CT-26, meaning there was more interaction with negatively charged cells. Moreover, the CBF-EGCG NPs showed antibacterial effects against *E. coli* and *S. aureus* and lipopolysaccharide (LPS) absorption, which help LPS clearance in damaged colon tissues. Positively charged CBF is more effective against Gram-negative bacteria because of electrostatic interaction with the LPS of bacteria. However, it is relatively less effective against Gram-positive bacteria, which CBF-EGCG NPs were able to compensate for with the ability of EGCG to inhibit peptidoglycan synthesis. Furthermore, the therapeutic effects following orally administered hydrogels included penetration of mucus and accumulation in colon tissues, as well as regulation of inflammation and the intestinal microbiome.

Caselli et al. investigated the amphiphilicity of a tryptophan (W)-end tagged KYE21 AMP and the performance of AMP-coated titanium oxide (TiO_2_) NPs [119]. The particle size of TiO_2_ decreased as the concentration of peptide used in the coating increased. KYE21 and WWWKYE21 coatings were stable during UV illumination, which produced reactive oxygen species by photocatalysis of TiO_2_ NPs. In addition, the WWWKYE21-coated TiO_2_ NPs interacted and disrupted bacterial- and lipopolysaccharide-like membranes and demonstrated increased effects with UV illumination. These results were the same for *E. coli*. Furthermore, cytotoxicity to human monocytes was low even after peptide coating. These effects resulted from the amphiphilicity of the W-end tagged peptide, which also contributed to the selectivity and antibacterial effects of the peptide-coated nanoparticles. W-tagging of AMPs played a role in strongly binding to the bacterial membrane where the W residue is negatively charged, thereby contributing to more effectively inducing the functions of AMPs and photocatalytic NPs.

Teng et al. introduced the notion that water-soluble AMP monomers with tryptophan (W), arginine (R), and tyrosine (Y), WRWRWRY, could form self-assembly by tyrosinase-induced oxidation at the wound site [120]. WRWRWRY maintained a soluble form without aggregates in water for 7 days, while oxidative WRWRWRY (mWRWRWRY) was self-assembled and showed a critical micellization concentration of 0.029 μg/mL. The self-assembly impacted the change in surface charge from +8 mV (WRWRWRY) to +37 mV (mWRWRWRY), which is an advantage in interacting with negatively charged bacterial membranes. The mWRWRWRY NPs could destabilize the membranes of *E. coli* and *S. aureus*, resulting in the MIC decreasing 4- to 7-fold compared to AMP monomers. Furthermore, the self-assembled NP formation demonstrated antioxidant activity compared to monomers with no activity. The skin wound recovery of S. aureus-infected mice treated with mWRWRWRY was faster than that treated with WRWRWRY.

The use of antimicrobial peptides (AMPs) in particle-based applications has emerged as a promising approach to enhance antimicrobial efficacy. Key trends include the encapsulation of AMPs in nanoparticles, such as liposomes and polymeric nanoparticles, which improves their stability, controlled release, and solubility. Functionalization of these nanoparticles with AMPs enables the development of antimicrobial coatings for medical devices and environmental surfaces, while targeted delivery systems using conjugated ligands or antibodies enhance specificity. Self-assembling nanostructures and combinations with other antimicrobial agents are also being explored to improve performance and address resistance. However, several limitations impact the widespread application of AMP-based nanoparticles. Safety concerns regarding potential toxicity to human cells and environmental impact require thorough evaluation. The complexity and cost of nanoparticle synthesis and functionalization pose significant challenges to commercial scalability. Additionally, issues related to the stability of nanoparticles and regulatory hurdles further complicate their development and approval.

In particle-based applications, the combination of AMPs with nanoparticles enhances antimicrobial efficacy by overcoming antibiotic resistance and providing synergistic effects. AMP–nanoparticle conjugates, such as peptide–gold nanoparticles, offer rapid bacterial killing and stable, controlled release, improving the overall performance of AMPs. These conjugates are particularly effective due to increased AMP density and surface area, enabling better interaction with bacterial membranes. However, concerns about nanoparticle-related toxicity, especially with smaller particles, pose significant challenges. The complexity and cost of nanoparticle synthesis and functionalization also hinder commercial scalability, while stability issues and regulatory hurdles further complicate their development.

**Table 3 antibiotics-13-00794-t003:** Biomedical applications of nanoparticles with AMPs.

AMP	Materials	Biomedical Results	Ref.
Ura56	Gold nanoparticle	Peptide stability against protease Bacteria-killing effect against antibiotic-resistant bacteria by membrane attachment and lysis Antibiofilm activity	[117]
LL-37	Titanium dioxide nanoparticle	Higher membrane attachment ability to anionic membrane compared to mammalian cell-like zwitterionic membrane	[121]
AS-48	Biomimetic magnetic nanoparticle	Enhanced growth inhibition effects against *E. coli* compared to free peptide	[122]
Ib-M2	Iron oxide nanoparticle coated with chitosan	Enhanced growth inhibiton effects against *E. coli* compared to free peptide	[123]
KYE21 and WWWKYE21	Titanium dioxide nanoparticle	Bacteria- and lipopolysaccharide-like membrane attachment using peptide Enhanced antibacterial effects against *E. coli* Selective toxicity between bacteria and human cell	[119]
NGIVKAGPAIAVLGEAAL and JH8194 sequence	Silver nanoparticle with silk fidronin	Silver release at pH 5.0 Bacterial membrane permeability and bactericidal effect against MRSA In vitro and in vivo osteogenic activity	[124]
LL37	Gold nanoparticle	Antibacterial effects against Gram-positive and Gram-negative bactria More or less cytotoxicity to endothelial cell Angiogenic activity	[125]
CCLLLLRRRRRR	Silver nanoclusters	Interaction with bacterial membrane targeting lipopolysaccharide 100-fold higher inhibition activity against *E. coli* compared to commercial silver nanoparticle	[126]
Thiol-terminated DDL_90_ BLG_10_	Gold nanoparticle	Antibacterial activity against MRSA using over-production of reactive oxygen species In vitro and in vivo biocompatibility	[127]
Cathelicidin-BF	Nanoparticles composed of epigallocatechin-3-gallate and silk fibroin	Increased antibacterial effects against *E. coli* Lipopolysaccaride adsorption In vivo therapeutic effect against ulcerative colitis	[118]
SAAP-148	Poly(lactic-co-glycolic) acid	Increased antibacterial activities (10–20 fold) and antibiofilm activities against antimicrobial-resistant *S. aureus* and *Acinetobacter baumannii*	[128]
PA-13	Chitosan Dextran sulfate	Improved stability against protease Maintained antibaterial activites against *Pseudomonas aeruginosa* in trysine-challenged conditions	[129]
SET-M33	poly(lactide-co-glycolide) conjugated with polyethylene glycol	Enhanced penetration of artifical mucus and bacterial alginate by PEGlyation Sustained release and persistent antibacterial activity against *Pseudomonas aeruginosa*	[130]
Trp-Arg-Trp-Arg-Trp-Tyr (WRWRWY)	Self-assembly after oxidization by tyrosinase	Positively increased surface charge Reactive oxygen stress scattering effects Stronger antibacterial efffect against *E.coli* and *S. aureus* compared to WRWRWY Boosted wound healing in mice skin	[120]
Gramicidin A‘ Alamethicin Melittin Indolicidin Pexiganan Cecropin A	lipid-based inverse bicontinuous cubic phase nanoparticles (cubosomes)	Enhanced antibacterial activity of indolicidin (against *S. aureus* and *Bacillus cereus*) and alamethicin (against *Bacillus cereus*) after cubosome formulation	[131]

*Escherichia coli* (*E. coli*). *Staphylococcus aureus* (*S. aureus*). Methicillin-resistant *Staphylococcus aureus* (MRSA).

### 4.3. Three-Dimensional Printing-Based Applications

Three-dimensional printing has emerged as a transformative technology in various biomedical applications, offering unprecedented precision and customization in the fabrication of biomedical devices and structures [132]. One of the promising areas of this technology is its integration with AMPs, which are short proteins capable of destroying bacteria, fungi, and viruses [133]; 3D-printed scaffolds are customized using surface modifications to achieve antimicrobial capabilities [134]. Since coatings and surface modifications have already been discussed in previous sections, the various 3D printing and design strategies used to develop antimicrobial 3D-printed scaffolds are discussed here (Table 4).

Ch et al. investigated the preparation and 3D printing of mucoadhesive gelatin methacryloyl (GelMA)/chitosan methacryloyl (ChiMA) hydrogels, fabricating them into contact lens-like patches (CLP) loaded with the AMP S100A12 for the treatment of bacterial keratitis (BK) [135]. Gelatin methacrylate (GelMA) and chitosan methacrylate (ChiMA) are polymers that confer mucoadhesive properties to the corneal tissue while also providing excellent mechanical strength. Notably, chitosan, utilized in the synthesis of ChiMA, is recognized for its inherent antibacterial activity. These attributes render GelMA and ChiMA highly suitable candidates for applications in 3D printing. In vivo experiments in a *P. aeruginosa*-infected BK rabbit model demonstrated that treatment with AMP-loaded CLP significantly reduced the bacterial load, as evidenced by colony-forming unit (CFU) assays. This novel delivery system incorporating AMP shows considerable potential to address the challenges of multidrug resistance (MDR) in bacteria and to reduce the frequency of dosing required with conventional eye drops. The inclusion of chitosan in the formulation enhances the synergistic effect of AMP, effectively disrupting bacterial biofilms.

Kruse et al. reported a significant reduction in infection rates by the common pathogen *S. aureus* on 3D-printed polyaryl ether ketone (PAEK) polymer surfaces, achieving a 4-log reduction through the covalent bonding of the AMP Mel4 via plasma immersion ion implantation (PIII) treatment [136]. The use of PIII to create covalent linkages with a broad-spectrum AMP like Mel4 provides an effective strategy for reducing microbial colonization on polyether ether ketone (PEEK) and polyether ketone (PEK) surfaces. Importantly, this method does not inhibit the proliferation of osteoblastic cells, ensuring biocompatibility. The study demonstrates that the various surface morphologies generated by different 3D printing processes do not diminish the antimicrobial efficacy of the peptide. 

Ullah et al. have developed an antibacterial and biocompatible silica-silk fibroin (SF) gel-based ink through innovative yet straightforward chemical methods involving sol-gel and self-assembly processes [137]. The covalent attachment of the AMP to the SF matrix imparts the scaffolds with significant bactericidal efficacy against both Gram-positive and Gram-negative bacteria. SF inherently lacks the arginine-glycine-aspartic acid (RGD) integrin-binding peptide motif, which is essential for promoting cell adhesion. The bio-conjugation of SF with antimicrobial peptides (AMPs), specifically of the cecropin-melittin type (CM), which possess the RGD sequence within their structure (referred to as CM-RGD), has been achieved using robust chemical methods. This modification aims to enhance the cell adhesion properties of the SF biopolymer. At optimized silica concentrations, the presence of the CM-RGD sequence on the silica-SF scaffold confers enhanced antibacterial activity after brief periods of co-incubation with Gram-negative bacteria. Interestingly, scaffolds without CM-RGD modification also exhibited bactericidal properties, particularly against Gram-positive bacteria, and extended co-incubation periods resulted in bactericidal effects against Gram-negative bacteria as well. The optimized silica-SF-CM-RGD scaffolds demonstrated not only potent anti-infective properties but also low cytotoxicity, promoting the growth and proliferation of osteoblast cells.

Three-dimensional printing technology has revolutionized biomedical applications, enabling the creation of customized medical implants and prosthetics embedded with AMPs, which enhance biocompatibility and reduce infection risks. This technology allows for the precise fabrication of AMP-loaded scaffolds for controlled drug delivery and the production of personalized wound dressings. The integration of AMPs in 3D-printed structures also offers continuous antimicrobial protection in various settings. However, compatibility issues between AMPs and 3D printing materials can affect peptide stability and activity. Additionally, the high costs, scalability challenges, and regulatory concerns, including safety standards and potential toxicity, remain significant obstacles to the widespread use of AMP-integrated 3D-printed products.

Recent developments include the creation of customized medical implants and prosthetics embedded with AMPs, which enhance biocompatibility and reduce infection risks. Additionally, 3D printing is used to fabricate personalized wound dressings with AMPs for improved fit and localized antimicrobial protection. The technology also enables the production of AMP-loaded scaffolds for controlled drug delivery and “smart” materials that release AMPs in response to specific stimuli. Functionalized surfaces with integrated AMPs are being developed for continuous antimicrobial activity in various settings. Challenges include the compatibility of AMPs with 3D printing materials, which can impact peptide stability and activity, and processing limitations of certain 3D printing methods. High costs and scalability issues also pose significant barriers. Regulatory and safety concerns, including compliance with safety standards and potential toxicity, require thorough evaluation. Additionally, long-term stability and performance of AMP-integrated 3D-printed products need further validation. In a comprehensive comparison, surface-based and nanoparticle-based applications demonstrate particular strengths in antimicrobial efficacy, especially in providing effective defense on specific surfaces or target sites. While nanoparticle-based and 3D printing applications offer high stability, they face significant challenges related to cost and commercial scalability, which require further innovations to overcome. Additionally, 3D printing-based applications show the greatest potential for customization and the development of innovative therapeutic strategies. However, technical limitations, cost, and safety concerns remain critical issues that must be addressed.

**Table 4 antibiotics-13-00794-t004:** Biomedical applications of 3D-printed technologies.

AMP	Materials	Biomedical Results	Ref.
S100A12	Mucoadhesive helatin methacryloyl/chisosan methacryloyl hedrogel	Strong antibacterial properties, reduced the bacterial load in vivo	[135]
Silk fibroin	Silica-silk fibroin-cecropin melittin-RGD aeregel	Potent bactericidal efficiency toward Gram-positive and Gram-negative bacteria, osteoconductivity of the scaffold.	[137]
Ponericin G1	BMP-2, poly(L-lactide-co-glycolide), dopamine	Maintain long-term antibacterial activity, cell adhesion, proliferation, and differentiation	[138]
3-poly-L-lysine (EPL)	polycaprolactone/ hydroxyapatite (PCL/HA)	Cytocompatible as well as capable of osteogenic differentiation and antimicrobial activity in vitro	[139]
Ponericin G1	gelatin/nanohydroxyapatite, dopamine	Both Gram-positive and Gram-negative bacteria (*E. coli* and *S. aureus*) were effectively inhibited up to 24 h, and the inhibition zone could remain for 72 h.	[140]
P1 (poly(L-lysine)), P2 (poly(Lglutamic acid))	N-carboxyanhydride (NCA) monomers	Antimicrobial displaying a significant 6–7-fold log10 reduction, with the built-in capacity to enhance the mechanical and biological properties	[141]
Mel4	Polyaryl ether ketone (PAEK)	reducing the microbial count on PEEK surfaces, no growth-inhibiting effect on osteoblastic cells	[136]
RWRWRWA-(Bpa)	Ultrafiltration membranes	cell membrane disruption, antibacterial activity and reduced biofilm growth	[142]

## 5. Challenges and Future Perspectives

AMPs exhibit both antibacterial and immune-modulatory activities, rendering them less susceptible to the development of resistance by bacterial pathogens. These advantages confer broad application prospects for peptides, although challenges related to the practical application of AMPs persist [3,6,9]. Despite their various benefits, AMPs are associated with certain limitations, such as toxicity, sourcing issues, short half-lives, large-scale production challenges, high production costs, and low permeability (Figure 5). These challenges have proven to be significant obstacles in the development of AMPs as viable alternatives to traditional antibiotics [3,52,97,143]. Continued research is necessary to address these limitations and fully realize the potential of AMPs in clinical settings. Strategies to mitigate toxicity, improve stability and half-life, enhance production methods, and reduce costs are essential. Moreover, advancements in delivery mechanisms could potentially overcome permeability issues, thus broadening the therapeutic application of AMPs. The ongoing exploration and optimization of these peptides will be crucial to addressing growing concerns over the increased prevalence of antibiotic resistance and the lack of development of novel antimicrobial therapies.

Recent studies have highlighted several key limitations of AMPs, two of which stand out: poor stability and susceptibility to proteolytic degradation. These limitations can result in a reduced half-life and restrict the range of possible administration routes for these drugs when targeting pathogens [52,143,144]. The stability of AMPs in vivo is generally low because they are easily degraded by both the endogenous digestive enzymes of the host and the enzymes secreted by pathogenic microorganisms [6,53]. To address these challenges, researchers have proposed various strategies to enhance the stability and bioavailability of AMPs in vivo. One approach involves synthesizing AMPs using D-type non-natural amino acids, which are less recognizable and thus less degradable by proteolytic enzymes. Another strategy is modifying the terminal ends of the AMPs through acetylation and amidation, processes that can shield peptides from enzymatic degradation. Additionally, the cyclization of AMPs has been explored to increase their structural stability, thereby reducing their susceptibility to proteolysis [145,146]. These strategies aim to extend the half-life of AMPs, allowing for more effective and flexible therapeutic applications. Continued research and development in these areas are essential for overcoming the current limitations and fully leveraging the potential of AMPs as powerful alternatives to conventional antibiotics. Advanced formulation techniques and novel delivery systems are also being investigated to further enhance the stability, efficacy, and clinical applicability of AMPs in the fight against antibiotic-resistant pathogens. 

Another important point is that large-scale production, and the high cost of that production, limit the use of AMPs [147]. Furthermore, only a few substances have been approved by regulatory bodies to date. Approved substances include glycopeptides such as vancomycin and teicoplanin, daptomycin, and polymyxin B. However, many AMPs are currently under investigation and testing for potential use [48]. Although numerous strategies have been developed to increase the stability of AMPs, these strategies have also been shown to decrease their antimicrobial activity [148].

Through the development of novel biomaterials and printing techniques, AMPs can be integrated with 3D printing technology to advance biomedical applications by creating smart, stimuli-responsive systems and personalized medical devices [133,149,150]. Future research may focus on hybrid drug delivery systems, scalable production methods, and the establishment of regulatory standards. Interdisciplinary collaboration and extensive clinical trials will be crucial for driving innovation, improving infection control, and enhancing personalized medicine and regenerative therapies [150,151,152,153]. 

Improvements in protective methods for AMP stability, cost-effective and scalable 3D printing techniques, clear regulatory frameworks, resistance monitoring, and optimization of the mechanical and functional properties of 3D-printed devices are necessary to facilitate the broader adoption and effectiveness of AMP-integrated 3D-printed biomedical applications. Several strategies are currently under development to counteract the above AMP limitations, and different design strategies such as sequence modification, periodization, and peptidomimetics have been used so far to improve proteolytic stability [95,154,155,156]. As one example, chemical modification strategies have been shown to be the most frequent and easiest way to improve AMP activity and biocompatibility [157,158]. The future use of AMPs is promising [3,159,160], and as research progresses, so does our ability to increase their bioavailability and efficacy, improve production efficiency, and lower production costs to enable larger-scale production [161]. It is anticipated that the next decade will witness significant advances in overcoming many of the limitations described here, with more successful case studies emerging at the clinical trial level and development reaching an industrially viable scale.

## 6. Conclusions

Antimicrobial peptides (AMPs) are emerging as potential substitutes for antibiotics, given their beneficial biological effects in host organisms and the challenge of resistance gene development. However, limitations remain in the production and application of AMPs across various fields and commercial uses. To enhance peptide yields, advanced genetic engineering and purification techniques should be further refined. Looking ahead, multifunctional and stimulus-responsive coatings are poised to play a pivotal role in future applications. Overcoming antimicrobial resistance and achieving stable antimicrobial performance requires more than just antimicrobial properties, and integrating surface modification technologies and antimicrobials is critical to achieving ideal antimicrobial utilization in biomedical applications. Ongoing research efforts are expected to lead to promising alternatives to conventional antibiotics to combat life-threatening and untreatable infections.

## Figures and Tables

**Figure 1 antibiotics-13-00794-f001:**
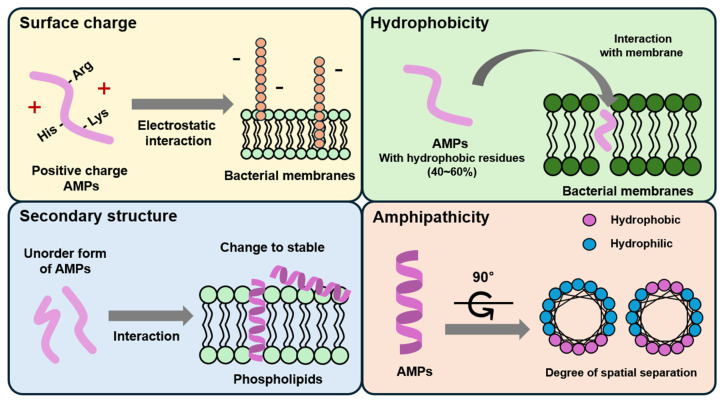
Features of peptides to reveal antibacterial effects. This figure explains design principles of antimicrobial peptides (AMPs) with four key categories, including positive surface charge, hydrophobicity, secondary structures, and amphipathicity.

**Figure 2 antibiotics-13-00794-f002:**
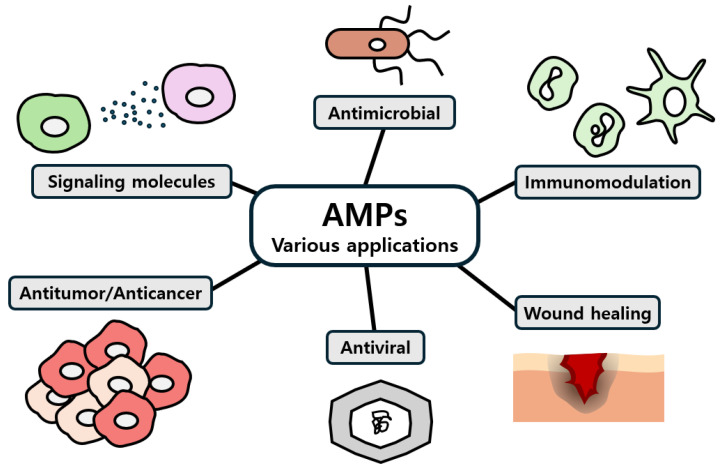
Various applications of antimicrobial peptides (AMPs). AMPs serve multiple biological functions, including antimicrobial activity, immunomodulation, antiviral properties, wound healing, antitumor/anticancer effects, and acting as signaling molecules in diverse physiological processes.

**Figure 3 antibiotics-13-00794-f003:**
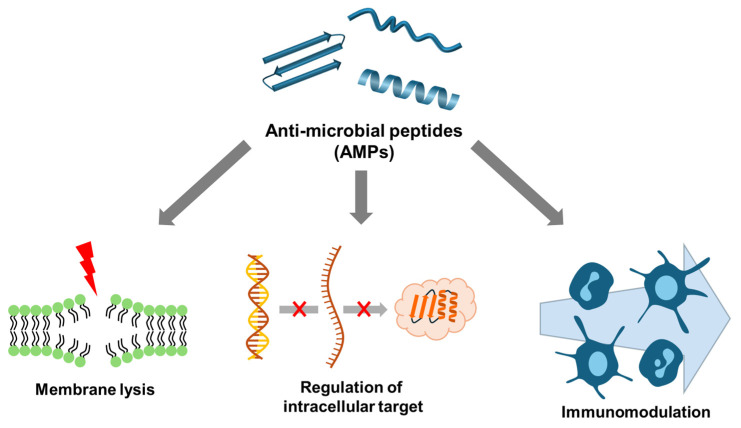
Mechanisms of action of antimicrobial peptides (AMPs). AMPs exert their effects through various pathways, including membrane lysis, inhibition of biofilm formation, regulation of intracellular targets, and modulation of the immune response [51].

**Figure 5 antibiotics-13-00794-f005:**
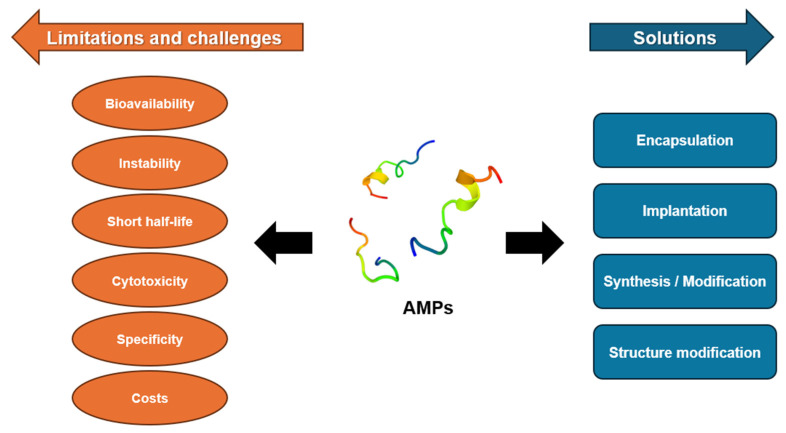
Schematic illustration of AMPs’ various limitations and possible solutions toward developing AMP-based approaches for biomedical applications [96,97].

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
