# Peer review of "Antimicrobial Peptides and Their Biomedical Applications: A Review"

_antibiotics, 2024, doi:10.3390/antibiotics13090794_

Round 1

Reviewer 1 Report

Comments and Suggestions for Authors

This review paper provides an interesting look at the application of AMPs beyond their well-known antimicrobial activities. While the paper looks good and is easily understandable, this reviewer has a few minor suggestions that could improve the paper.

1. In page 1, lines 34 to 36, the authors mentioned that AMPs are resilient to extreme pH and temperature. I suggest that the authors cite some references to provide examples of AMPs of this nature.

2. There seems to be a reference in Figure 3 that was not cited. Brackets are found in Line 148.

3. Sections 3 and 4 summarizes modification of AMPs by bioconjugation with biomaterials and various roles that AMPs play in different applications. However, there should be inferences from the authors as to how these works, particularly in Section 3. For example, what could be the role of mPEG in the CT9W1000 in enhancing the antibacterial effects of T9W, among the other effects of PEGylation? The same question can be applied to the other examples provided by the authors.

In both sections, what are the different limitations of these applications?

4. The authors made good points in presenting the key limitations of AMPs, in general. However, it would be interesting to know, within the context of this paper, whether the limitations may be overcome by the previously discussed AMP modifications.

Author Response

This review paper provides an interesting look at the application of AMPs beyond their well-known antimicrobial activities. While the paper looks good and is easily understandable, this reviewer has a few minor suggestions that could improve the paper.

  1. In page 1, lines 34 to 36, the authors mentioned that AMPs are resilient to extreme pH and temperature. I suggest that the authors cite some references to provide examples of AMPs of this nature.

- Answer: Thanks for your comments. We added some references in page 1, lines 39 to 40.

  1. There seems to be a reference in Figure 3 that was not cited. Brackets are found in Line 148.

- Answer: Thanks for your comments. We added reference in Figure 3.

  1. Sections 3 and 4 summarizes modification of AMPs by bioconjugation with biomaterials and various roles that AMPs play in different applications. However, there should be inferences from the authors as to how these works, particularly in Section 3. For example, what could be the role of mPEG in the CT9W1000 in enhancing the antibacterial effects of T9W, among the other effects of PEGylation? The same question can be applied to the other examples provided by the authors.

In both sections, what are the different limitations of these applications?

- Answer: Thanks for your comments. We agree with your comment. We added contents of you mentioned. Also, at the request of reviewers, a discussion section was written for each session.

  1. The authors made good points in presenting the key limitations of AMPs, in general. However, it would be interesting to know, within the context of this paper, whether the limitations may be overcome by the previously discussed AMP modifications.

- Answer: Thanks for your comments. We agree with your comment. We added the discussion in section. Also, at the request of reviewers, a discussion section was written for each session.

Reviewer 2 Report

Comments and Suggestions for Authors

Comments and suggestions to the authors are attached as pdf.

Author Response

Here are a few comments to be addressed:

  1. It would be great if authors could include a section immediately after introduction or at least a full paragraph on what are the essential chemical and structural prerequisites for a peptide molecule to be antimicrobial peptide? Be good to showcase basic design principles involved in terms of what chemical residues/moieties are essential etc. I do believe that providing such fundamental aspects would help readers with basic platform and knowledge to understand and relate well to the remaining length and aspects discussed in the later sections and makes this review article wholesome.

- Answer: Thanks for your comments. We agree with your comment. We added contents in introduction section, second paragraph.

  1. Lines 32-34, AMPs are widely found in nature and play a crucial role in the innate immune systems of various organisms, exhibiting broad inhibitory effects against bacteria, fungi, parasites, and viruses. Authors need to provide reference/s for this statement.

- Answer: Thanks for your comments. We added some references in lines 37 to 39.

  1. Line 71, dominant residues As per authors what does “dominant residues” refer to?

- Answer: Based on the reference "Truncated and constrained helical analogs of antimicrobial esculentin-2EM," the dominant residues refer to cationic residues and hydrophobic residues. Both helicity and amphipathicity are considered as to be essential characteristics of this class of AMPs; the hydrophobic face is important for the interaction with the hydrophobic interior of the cell membrane, whereas the hydrophilic face plays a critical role in specific interactions with the membrane surfaces of certain microbes.

We edited sentence in lines 95 to 98. The sentence has been clearly revised.

  1. Lines 105-107, positively charged and amphiphilic AMPs bind to negatively charged and

hydrophobic phospholipids in the cell membrane. This gives an impression to the reader that phospholipids are hydrophobic by nature. This line should be clearly written as positively charged and amphiphilic AMPs bind to negatively charged amphipathic phospholipids in the cell membrane.

- Answer: Thanks for your comments. We agree with your comment. We edited sentence in line 138 to 140.

  1. Line 148 Figure 3, in the caption [] is a reference missing there?

- Answer: Thanks for your comments. We made mistakes. We added reference in figure 3.

  1. Authors surely did an impressive job. However, it would be nice if authors could correlate the findings and bring a rational or scientific discussion in best possible way toward end of each section. I hardly noticed any such discussion. In absence of such valuable discussions the article would just be only a compilation of some findings from a bunch of publications!

- Answer: Thanks for your comments. We agree with your comment. We've included an insightful discussion of the content covered in each session.

Reviewer 3 Report

Comments and Suggestions for Authors

This review summarizes the various strategies used to tackle antibiotic resistance, lists Most of the recent studies on the properties and impregnation methods of AMPs, and provides researchers with insights into the latest advancements in AMP research. However, there are still some major issues to address before publication.

1.     The title of this review is "Antimicrobial Peptides and Their Biomedical Applications." Therefore, the focus of this review should be on providing an overview of antimicrobial peptides, including their introduction, development, antimicrobial mechanisms first. Then, the biomedical applications of antimicrobial peptides in various fields should be discussed, as illustrated in Figure 1. However, the second half of this review does not follow the framework presented in Figure 1 to review the biomedical applications of antimicrobial peptides. This results in a lack of clear logical structure in the review. The authors are advised to make the necessary adjustments.

2.     The expression 'Various AMP Strategies' in the title of Section 4 of this review may not be appropriate. Based on the authors' description, this section should focus on strategies for enhancing the efficacy, stability, and delivery of antimicrobial peptides, which cannot be broadly categorized as 'AMP Strategies.' The authors should adjust this expression accordingly.

3.     In Section 4 of this review, the authors introduce the applications of antimicrobial peptides through Surface-Based Applications, Particle-Based Applications, and 3D Printing-Based Applications. However, these three aspects are essentially all about coating, and modifying antimicrobial peptides on material surfaces. The only difference lies in the three different types of materials, rather than discussing the construction strategies of antimicrobial peptide materials.

4.     In Section 3 of this review, the introduction of enhancing antimicrobial activity through constructing different antimicrobial peptide-modified materials lacks a clear connection to the overall context of the review, resulting in an unclear overall structure.

5.     The title of this review is 'Antimicrobial Peptides and Their Biomedical Applications.' However, the review does not classify and discuss the antimicrobial applications of antimicrobial peptides in different biomedical fields, nor does it address the advantages and disadvantages of their applications in these fields. The authors are advised to make the necessary revisions

Comments on the Quality of English Language

Some expressions are not  appropriate.

Author Response

This review summarizes the various strategies used to tackle antibiotic resistance, lists Most of the recent studies on the properties and impregnation methods of AMPs, and provides researchers with insights into the latest advancements in AMP research. However, there are still some major issues to address before publication.

  1. The title of this review is "Antimicrobial Peptides and Their Biomedical Applications." Therefore, the focus of this review should be on providing an overview of antimicrobial peptides, including their introduction, development, antimicrobial mechanisms first. Then, the biomedical applications of antimicrobial peptides in various fields should be discussed, as illustrated in Figure 1. However, the second half of this review does not follow the framework presented in Figure 1 to review the biomedical applications of antimicrobial peptides. This results in a lack of clear logical structure in the review. The authors are advised to make the necessary adjustments.

- Answer: Thanks for your comments. We revised the manuscript as reviewers’ comments and requests. We have supplemented the contents.

  1. The expression 'Various AMP Strategies' in the title of Section 4 of this review may not be appropriate. Based on the authors' description, this section should focus on strategies for enhancing the efficacy, stability, and delivery of antimicrobial peptides, which cannot be broadly categorized as 'AMP Strategies.' The authors should adjust this expression accordingly.

- Answer: Thanks for your comments. We revised the title of section 4 to “AMP Application for Biomaterials” to better reflect the overall content of section 4.

  1. In Section 4 of this review, the authors introduce the applications of antimicrobial peptides through Surface-Based Applications, Particle-Based Applications, and 3D Printing-Based Applications. However, these three aspects are essentially all about coating, and modifying antimicrobial peptides on material surfaces. The only difference lies in the three different types of materials, rather than discussing the construction strategies of antimicrobial peptide materials.

Answer: Thanks for your comments. We added "biomaterials" and deleted "strategy" to express section 4 of the appropriate title. Also, we modified the contents and table caption to introduce diverse materials. At the request of reviewers, we’ve included a discussion of the content covered in each session.

  1. In Section 3 of this review, the introduction of enhancing antimicrobial activity through constructing different antimicrobial peptide-modified materials lacks a clear connection to the overall context of the review, resulting in an unclear overall structure.

- Answer: Thanks for your comments. We revised the expression. Section 3 of this review has been enhanced to ensure clear and cohesive connections with the overall context of the review.

  1. The title of this review is 'Antimicrobial Peptides and Their Biomedical Applications.' However, the review does not classify and discuss the antimicrobial applications of antimicrobial peptides in different biomedical fields, nor does it address the advantages and disadvantages of their applications in these fields. The authors are advised to make the necessary revisions

- Answer: Thanks for your comments. We agree with your comment. At the request of reviewers, we've included an insightful discussion of the content covered in each session. So, we correlated the findings and bring a rational or scientific discussion in toward end of each section.

Reviewer 4 Report

Comments and Suggestions for Authors

General comments:

The current review is well-written, and the flow is coherent. It discusses the use of AMPs in different biomedical applications. The authors explain the most recent advances in engendering AMPs, delivery methods, and their combination with biomaterials. They also discuss the major current issues, advantages, and perspectives on AMPs applications.

Minor comments:

-Figure 2 explains the roles of AMPs; however, it looks more like a summary of the action´s mode. Membrane lysis, regulation of intracellular targets, and anti-biofilms have the role of killing pathogens. It would be nice for Figure 2 to summarize the action mode of AMPs, including membrane lysis and regulation of intracellular targets, as is already in the figure. However, anti-biofilm is the consequence of a mechanism; explain the mechanism.

-In addition, immunomodulation is a consequence of AMPs' action, which can also regulate many functions in a similar way to immunity. It would be nice if you could include the mechanisms by which AMPs exert the immunomodulatory effect instead of only mentioning immunomodulation. For example, they play such a role because they work as ligands, or inhibitors of receptors, etc. This also must be discussed in the text in section 2.2. Non-Membrane Targeting, in the same nice way, as is discussed that AMPs inhibit transcription and translation.

-Figure legends could also be extended to explain better what the authors want to show.

Comments on the Quality of English Language

The authors use the word AMP instead of AMPs in several lines, for example, Line 187, Figure 1. Just for clarity, the authors should correct it.

Author Response

General comments:

  1. The current review is well-written, and the flow is coherent. It discusses the use of AMPs in different biomedical applications. The authors explain the most recent advances in engendering AMPs, delivery methods, and their combination with biomaterials. They also discuss the major current issues, advantages, and perspectives on AMPs applications.

 - Answer: Thanks for your comments.

Minor comments:

  1. Figure 2 explains the roles of AMPs; however, it looks more like a summary of the action´s mode. Membrane lysis, regulation of intracellular targets, and anti-biofilms have the role of killing pathogens. It would be nice for Figure 2 to summarize the action mode of AMPs, including membrane lysis and regulation of intracellular targets, as is already in the figure. However, anti-biofilm is the consequence of a mechanism; explain the mechanism.

- Answer: Thanks for your comments. We have supplemented the action mode of AMPs, including membrane lysis and regulation of intracellular targets, as is already in the figure. Also, we revised the figure 2 and focused on the membrane lysis, regulation of intracellular targets, and immunomodulation. Of course, the mechanism of anti-biofilm and explanation of AMPs were described.

  1. In addition, immunomodulation is a consequence of AMPs' action, which can also regulate many functions in a similar way to immunity. It would be nice if you could include the mechanisms by which AMPs exert the immunomodulatory effect instead of only mentioning immunomodulation. For example, they play such a role because they work as ligands, or inhibitors of receptors, etc. This also must be discussed in the text in section 2.2. Non-Membrane Targeting, in the same nice way, as is discussed that AMPs inhibit transcription and translation.

- Answer: We agree with your comment. We made mistakes. We edited sentence in section 2.2, line 218 to 229.

  1. Figure legends could also be extended to explain better what the authors want to show.

- Answer: We agree with your comment. We edited legends to clarify them.

  1. The authors use the word AMP instead of AMPs in several lines, for example, Line 187, Figure 1. Just for clarity, the authors should correct it.

- Answer: Thanks for your comments. We made mistakes and have edited all the text in the manuscript.

Reviewer 5 Report

Comments and Suggestions for Authors

The manuscript "Antimicrobial Peptides and Their Biomedical Applications: A Review" by Kim, Ki, and Pack provides an excellent overview of the research on AMPs and their applications.

The review is well-organized with proper figures and tables, covering AMP mechanisms of action, strategies for application, and challenges and future outlooks. This manuscript is very helpful for the current research and industry. However, the discussion on "immunomodulatory activity" is somewhat lacking, and the nanoparticle section is more focused on metal nanoparticles. It would benefit from including discussions on other organic nanoparticles which is popular in the current pharmaceutical industry, e.g., liposomes and hydrogels, to offer a more balanced view.

I have also included some minor suggestions for your reference. However, these will not affect the overall quality of the review:

- Line 148: A reference is missing.

- In Figure 3, the detergent-like illustration is a bit small.

- Table 1: "Amidoamine" is typically written as one word, so no space is needed.

- The leftmost parts of lines 186-189 and 229-231 are not aligned.

- Line 193 and many other instances: Generally, "last name + et al." should not be italicized

- The reference list may use a mix of article titles in the Capitalisation of Each Word and plain words, and a mix of journal names in abbreviations and full names, e.g., lines 696, 699, 750, 763, and 770. 

- Line 760. I believe the journal name is "Pharmaceuticals-Basel" not "Pharmaceuticals-Base". Please check your reference list before final submission. 

Thank you for your review of this topic, and I look forward to seeing more of your excellent articles in the future.

Author Response

The manuscript "Antimicrobial Peptides and Their Biomedical Applications: A Review" by Kim, Ki, and Pack provides an excellent overview of the research on AMPs and their applications.

  1. The review is well-organized with proper figures and tables, covering AMP mechanisms of action, strategies for application, and challenges and future outlooks. This manuscript is very helpful for the current research and industry. However, the discussion on "immunomodulatory activity" is somewhat lacking, and the nanoparticle section is more focused on metal nanoparticles. It would benefit from including discussions on other organic nanoparticles which is popular in the current pharmaceutical industry, e.g., liposomes and hydrogels, to offer a more balanced view.

- Answer: Thanks for your comments. In section 2.2, the immunomodulation section was added to the last paragraph. In Section 4.2, the contents related to organic particles were added to the table and added to the main text.  

I have also included some minor suggestions for your reference. However, these will not affect the overall quality of the review:

  1. Line 148: A reference is missing.

- Answer: Thanks for your comments. We made mistakes. We added reference.

  1. In Figure 3, the detergent-like illustration is a bit small.

- Answer: Thanks for your comments. We edited illustration.

  1. Table 1: "Amidoamine" is typically written as one word, so no space is needed.

- Answer: Thanks for your comments. We edited it as one word.

  1. The leftmost parts of lines 186-189 and 229-231 are not aligned.

- Answer: Thanks for your comments. We edited it.

  1. Line 193 and many other instances: Generally, "last name + et al." should not be italicized

- Answer: Thanks for your comments. We edited it.

  1. The reference list may use a mix of article titles in the Capitalisation of Each Word and plain words, and a mix of journal names in abbreviations and full names, e.g., lines 696, 699, 750, 763, and 770.

- Answer: Thanks for your comments. We edited references.

  1. Line 760. I believe the journal name is "Pharmaceuticals-Basel" not "Pharmaceuticals-Base". Please check your reference list before final submission.

- Answer: Thanks for your comments. We checked again and edited references.

Thank you for your review of this topic, and I look forward to seeing more of your excellent articles in the future.

Round 2

Reviewer 2 Report

Comments and Suggestions for Authors

Please find an attachment for the comments

Author Response

After major revision, the review titled, “Antimicrobial Peptides and Their Biomedical Applications” has been tremendously improved.

Indeed, a very impressive job done by the authors by improving the text and adding some more

information by considering the comments. Here is a one last comment to be taken care.

- Answer: Thanks for your comments which are great helpful for improving our paper.

Comment:

However, I strongly recommend authors to add one representative figure or a chemdraw

schematic for the new discussion added on design principles discussed in the added paragraph

(lines 49-68), to make this review an outstanding and more useful to the broader community of

readers.

- Answer: Thanks for your comments. We agree with your comment. We added representative figure in section 1 (lines 49 to 53).

Reviewer 3 Report

Comments and Suggestions for Authors

The authors have responded to and revised the manuscript according to the first round of review comments. However, some minor issues remain unresolved. For instance, in response to comment 5, the authors only added examples of applications in different biomaterials but did not summarize and compare the advantages and disadvantages of antimicrobial peptides in various biomedical applications. Additionally, it is important that the authors indicate where in the manuscript each revision corresponding to the reviewer’s comments has been made. This is crucial for the review process.

Author Response

The authors have responded to and revised the manuscript according to the first round of review comments. However, some minor issues remain unresolved.

For instance, in response to comment 5, the authors only added examples of applications in different biomaterials but did not summarize and compare the advantages and disadvantages of antimicrobial peptides in various biomedical applications.

- Answer: Thanks for your comments. We agree with your comment. At the request of the reviewer, we summarized and compared the advantages and disadvantages of antimicrobial peptides in various biomedical applications at the end of each section (line 360-369, line 452-461, line 518-527, and line 540-548).

Additionally, it is important that the authors indicate where in the manuscript each revision corresponding to the reviewer’s comments has been made. This is crucial for the review process.

- Answer: We have written the revised contents in red to indicate where the changes have been made. We apologize for the shortcomings of the first version, in this regard. We will indicate where in the manuscript each revision corresponding to the reviewer’s comments has been made.